 eLife

 

# The pupillary light response as a physiological index of aphantasia, sensory and phenomenological imagery strength

Lachlan Kay[1†], Rebecca Keogh[1,2*†], Thomas Andrillon[1,3], Joel Pearson[1]

[1]School of Psychology, University of New South Wales, Sydney, Australia; [2]School of Psychological Sciences, Macquarie University, Sydney, Australia; [3]Sorbonne Université, Institut du Cerveau - Paris Brain Institute - ICM, Inserm, CNRS, Paris, France

**Abstract** The pupillary light response is an important automatic physiological response which optimizes light reaching the retina. Recent work has shown that the pupil also adjusts in response to illusory brightness and a range of cognitive functions, however, it remains unclear what exactly drives these endogenous changes. Here, we show that the imagery pupillary light response correlates with objective measures of sensory imagery strength. Further, the trial-by-trial phenomenological vividness of visual imagery is tracked by the imagery pupillary light response. We also demonstrated that a group of individuals without visual imagery (aphantasia) do not show any significant evidence of an imagery pupillary light response, however they do show perceptual pupil light responses and pupil dilation with larger cognitive load. Our results provide evidence that the pupillary light response indexes the sensory strength of visual imagery. This work also provides the first physiological validation of aphantasia.

## Editor's evaluation

This is a rigorous study of the relation between the vividness of visual imagery and the pupillary light response that can result from it. It provides evidence for the absence of imagery in individuals that self-report as aphantasic. The results will likely be of interest to researchers in a range of disciplines such as psychology, neuroscience and philosophy.

**\*For correspondence:**
rebeccalkeogh@gmail.com

†These authors contributed equally to this work

**Competing interest:** The authors declare that no competing interests exist.

## Introduction

Our pupil's ability to change size is an important physiological response that adjusts the amount of light hitting the retina to optimize vision and protect the retina. Pupils constrict in response to brightness whereas they dilate in response to dark conditions (known as the pupillary light response or reflex); while these responses are related, they are considered to be driven by different neural pathways (see *Mathôt, 2018* for a review). These involuntary pupil responses were once thought to be driven only by afferent visual stimulation, or automatic activation from emotional responses (*Bradley et al., 2008*; *Partala and Surakka, 2003*), however, recent studies suggest that pupil size is sensitive to higher order perceptual and cognitive processes. For example, subjective interpretation of equiluminant stimuli, such as greyscale images of the sun elicit greater pupil constriction than those of the moon (*Binda et al., 2013b*). The target of covert visual attention can drive pupillary light responses (*Binda et al., 2013a*), as can visual working memory content (*Zokaei et al., 2019*), but see *Blom et al., 2016*. Further, evidence suggests that it might be mental imagery that is driving some of these cognitively induced pupil responses (*Laeng and Sulutvedt, 2014*) and recent work has shown

that there are pupillary light responses even when reading or listening to words conveying some level of brightness (*Mathôt et al., 2017*). Hence, it remains unknown if the variations in pupil response to equiluminant stimuli are due to high-level semantic content or low-level visual imagery.

Visual imagery is considered a useful and often essential tool in many aspects of cognition. It plays an important role in the retrieval of items from short- and long-term memory (*Pearson, 2019*), visual working memory (*Keogh and Pearson, 2011*; *Keogh and Pearson, 2014*; *Pearson and Keogh, 2019*), acquisition of language (*Just et al., 2004*), and spatial navigation (*Sack et al., 2005*; *Guariglia and Pizzamiglio, 2007*). It is also used for simulating both past and potential future events (*Schacter et al., 2012*; *Schacter and Madore, 2016*), the latter often as a form of self-motivation for goal attainment (*Szpunar et al., 2007*). As essential to cognition as it might appear, large individual differences exist in visual imagery and its vividness. Some people report imagery as so vivid it feels almost like perception, while a small percentage of otherwise healthy people seemingly do not have the capacity for visual imagery at all – they report that when they think about how an object looks, there is no sensory-like experience of it whatsoever (*Galton, 1880*). This condition has been recently termed 'aphantasia' (*Zeman et al., 2015*); it can be congenital, persisting throughout one's lifetime (*Zeman et al., 2015*) or acquired (*Zeman et al., 2010*), is associated with a range of differences in general cognition (*Dawes et al., 2020*; *Keogh et al., 2021a*, *Keogh and Pearson, 2021*), including dampened fear responses to imagined scary scenarios (*Wicken et al., 2021*). The existence of aphantasia has also been established using objective techniques that measure the low-level sensory elements of imagery (*Keogh and Pearson, 2018*).

The rationale of the current study was to accurately and objectively utilize individual differences in mental imagery (both in the general population and aphantasia) to provide strong evidence that it is the sensory strength and subjective vividness of imagery that drives the cognitive pupillary light response. Similar rationale has been previously used by linking the vividness and objective sensory strength of imagery to behavioural or neurological measures (*Bergmann et al., 2016*; *Shine et al., 2015*; *Wassell et al., 2015*). If imagery plays a causal role in endogenous pupil size changes, then individual differences in imagery should be reflected in these measures.

Here, we utilized both subjective and objective measures of visual imagery ability and show that, within the same individual, greater pupillary light responses during imagery are associated with reports of stronger and more vivid imagery. We then used this task to compare imagery strength between individuals and test the veracity of the self-reported lack of imagery in aphantasia. We show that while aphantasic individuals display pupil contraction to perceptual brightness and dilation with effort (cognitive load), they do not show any evidence of pupil change in response to attempts at imagery – providing the first objective physiological evidence confirming the existence of aphantasia.

## Results

### The imagery pupillary light response in the general population

In the pupillometry imagery task (based on *Laeng and Sulutvedt, 2014*; see *Figure 1A*), participants who reported having visual imagery were presented with one or four 'Bright' or 'Dark' triangles for 5 s (see *Figure 1—figure supplement 1* for images used). Following this they viewed a blank screen for 8 s (which allowed any after-images to fade) and were then instructed to imagine the prior image/s for 6 s, after which they rated the vividness of their imagery from 1 to 4. Pupils showed a clear pupillary light response to perceptual images (*Figure 1B*; perception section; a significant effect of perceptual luminance $F(1, 41) = 190.02$, $p < 0.001$.) This trend was mirrored in the imagery period showing a significant main effect of imagery luminance (*Figure 1B*, box insets: imagery section; $F(1, 41) = 67.42$, $p < 0.0001$), indicating that imagery also demonstrates a pupillary light response. Post hoc analysis using the Bonferroni correction for multiple comparisons found that for both Set-Size-One and Set-Size-Four, the pupil size in the Dark condition was significantly greater than in the Light condition during imagery ($p < 0.001$ and $p < 0.05$, respectively, see *Figure 1C*). There was no main effect of set size during perception $F(1, 42) = 2.67$, $p = 0.11$. However, there was a significant main effect of set size during imagery $F(1, 41) = 6.48$, $p = 0.015$, with less constriction/more dilation for Set-Size-Four (when averaged across the brightness conditions). This is consistent with previous studies suggesting that pupil size is influenced by cognitive load (*Kahneman and Beatty, 1966*; *Laeng et al., 2011*; *van der Wel and van Steenbergen, 2018*). Post hoc analysis also demonstrated that in the Bright



**Figure 1.** Pupillary response task schematic and eye-tracker results for the general population. (**A**) Pupillometry imagery experiment timeline. Each trial began with the presentation of a white fixation cross at the centre of a grey screen (baseline screen) for 1 s. An image was then presented at the centre of this grey screen for 5 s (either one or four triangles of varying brightness, see *Figure 1—figure supplement 1* for illustrations of all stimuli). Participants were instructed to focus on the stimuli during this time and memorize its size, orientation, and level of brightness. Next, a black screen with

*Figure 1 continued on next page*

*Figure 1 continued*

a white fixation cross was presented for 8 s, allowing the perceived after-image to completely fade and pupils to dilate back to equivalent resting levels. The grey baseline screen was then presented again for 6 s. During this time, participants were cued (via two auditory beeps) to actively start imagining the stimuli observed previously during that trial, while maintaining focus on the fixation cross. These beeps were presented 1 s into the grey screen period leaving 5 s of imagery time. Lastly, participants were prompted to report the vividness of their imagery during those previous 5 s on a scale of 1–4 (1 being 'not vivid at all – no shape appeared in imagery'; 4 being 'very vivid – almost like seeing it') via key response. (**B**) Mean pupil size waveforms for the general population, presented as mm change from baseline. Left panel: data averaged across the course of a trial for Bright (red lines) and Dark (blue lines) conditions for the general population. *Right panels:* Set-Size-One and Set-Size-Four conditions are shown separately during the imagery period (i.e. pupil size from seconds 15 to 20). Shaded error bands represent the standard error of the mean (± standard error of the mean [SEM]). (**C**) Mean pupil size change from baseline during imagery (i.e. averaged from seconds 15 to 20 of trials) of Bright (red bars) and Dark stimuli (blue bars). (**D**) Pupil-difference scores (difference in pupil size during imagery between bright and dark conditions) as a function of subjective vividness ratings for Set-Size-One and Set-Size-Four conditions. Data points represent one participant. Error bars indicate ± SEM, calculated across participants. *p < 0.05, ***p < 0.0001.

The online version of this article includes the following source data, source code, and figure supplement(s) for figure 1:

**Source code 1.** r Code for LME analysis of vividness ratings.

**Source data 1.** Source data for *Figure 1C*.

**Source data 2.** Source data for *Figure 1D*.

**Figure supplement 1.** Stimuli used in the experiment.

**Figure supplement 2.** Correlations between pupil-difference scores and mean trial vividness.

**Figure supplement 3.** Correlations between pupil-difference scores and Vividness of Visual Imagery Questionnaire (VVIQ).

condition, Set-Size-Four resulted in significantly more pupil dilation during imagery than Set-Size-One (p = 0.001). However, in the Dark condition pupil dilation during Set-Size-Four imagery was not significantly different to Set-Size-One (p = 0.266).

Prior behavioural work suggests we have reasonable metacognition of visual imagery, that is we are able to estimate the strength of imagery on a trial-by-trial basis (*Pearson et al., 2011*; *Rademaker and Pearson, 2012*). Here, we compared pupil responses to the trial-by-trial ratings of vividness. Pupil-difference scores are shown as a function of intraindividual vividness ratings for Set-Size-One and Set-Size-Four (see *Figure 1D*). A 2 × 4 linear mixed-effects analysis (2 (set size: 1, 4) × 4 (vividness rating: 1, 2, 3, 4)) demonstrated there was a significant effect of vividness ($\chi^2(3) = 49.54$, p = 1.004e$^{-10}$), with larger pupillary light response for more vivid imagery trials (for both set sizes, see *Figure 1D* and fixed effects estimates in *Supplementary file 1*). These data demonstrate that the pupillary light response tracks the phenomenological vividness of visual imagery from moment to moment.

If the sensory strength of imagery is indeed driving the imagery pupillary light response, then the degree to which this response occurs should be related to independent objective measures of imagery strength in each individual. To assess this, we utilized the binocular rivalry method (*Pearson, 2014*; *Pearson et al., 2008*), which allows the objective assessment of the sensory strength of imagery, without relying on any subjective reports (*Chang and Pearson, 2018*). This is achieved by measuring the degree to which an individual's imagery biases subsequent binocular rivalry perception. We compared pupil-difference scores (imagery of dark stimuli–bright stimuli, such that larger scores indicate a larger pupillary light response) with imagery strength measured using the binocular rivalry paradigm, in which higher priming scores indicate stronger imagery (*Figure 2A*; *Pearson et al., 2008*; *Pearson et al., 2011*). Within the general population, degree of pupil change in the Set-Size-One condition correlated positively with imagery strength, using Pearson's correlation coefficient ($r_p(41) = 0.62$, p = <0.0001, see *Figure 2B*: green circles and green trendline). The Set-Size-Four pupil data set violated normality (Shapiro–Wilk test, p = 0.003), therefore, the Spearman's correlational coefficient was used to assess its relationship with binocular rivalry priming. A significant positive correlation was found between Set-Size-Four pupil-difference scores and binocular rivalry priming ($r_s(41) = 0.46$, p = 0.002, see *Figure 2C*: green circles and green trendline). This provides further evidence that the sensory strength of imagery content is driving the imagery pupillary light response.

## Aphantasia and the imagery pupillary light response

Our results indicate that the strength of the content of imagery drives the imagery pupillary light response in participants who experience visual imagery. The involuntary nature of this response

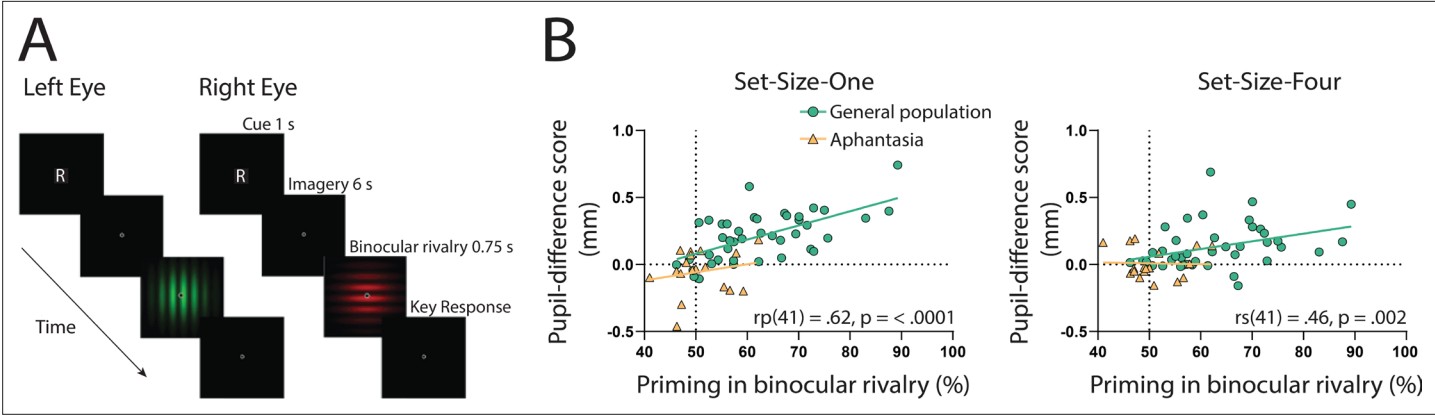

**Figure 2.** Binocular rivalry task schematic and correlational results. (**A**) Example of an imagery trial for the binocular rivalry paradigm. Participants were cued to imagine either a red or green Gabor pattern prior to binocular rivalry with the letter 'R' or 'G' (750 ms). Participants then imagined the image for 6 s, after which they were presented with the binocular rivalry display (750 ms) and were asked to indicate which image was dominant. Trials where participants reported seeing the pattern they were cued to imagine as dominant were denoted as 'primed' trials. The number of primed trials divided by the total number of trials (excluding mock trials and mixed percepts) was used to calculate a percent primed score for each participant. (**B**) Correlation between visual imagery strength, as measured by the pupillary response task (pupil-difference score: difference between bright and dark conditions) and visual imagery strength as measured by the binocular rivalry task. Set-Size-One (left) and Set-Size-Four (right) conditions are shown. Scatterplots show the general population (green circles and green trendline) and aphantasic individuals (yellow triangles and yellow trendline) data. Correlation coefficients refer to the general population only (green trendline). All data points represent one participant.

The online version of this article includes the following source data and figure supplement(s) for figure 2:

**Source data 1.** Source date for *Figure 2B*.

**Figure supplement 1.** Correlations between pupil-difference scores and VVIQ.

**Figure supplement 2.** Correlations between pupil-difference scores and binocular rivalry priming during perception.

**Figure supplement 3.** Correlations between mean eccentricity and pupil-difference scores during imagery for control and aphantasic groups.

**Figure supplement 4.** Correlations between mean eccentricity during imagery and binocular rivalry priming for control and aphantasic groups.

**Figure supplement 5.** Correlations between mean eccentricity during imagery and mean trial-by-trial vividness scores for the control group.

**Figure supplement 6.** Binocular rivalry and mock rivalry priming scores for control and aphantasic individuals.

**Figure supplement 7.** One-sample *t*-test prior and posterior plots.

**Figure supplement 8.** VVIQ scores for control and aphantasic individuals.

provides a valuable objective measure of imagery strength. Accordingly, we sought to utilize this finding to test the veracity of a condition called aphantasia, that is if these individuals truly lack visual imagery, they should not show a pupillary light response to imagined images. However, if aphantasic individuals do show an imagery-based pupillary light response, one might interpret this as a form of imagery existing, but below threshold for conscious phenomenological awareness. We ran this same study in 18 aphantasic participants and compared their performance to that of the general population. These participants had contacted the lab reporting their lack of visual imagery and asked to participate in our research. They were also unaware of the goals and hypotheses of the current study. Aphantasia was confirmed in these individuals using self-report questionnaires (Vividness of Visual Imagery Questionnaire [VVIQ] score <32) and by means of our binocular rivalry priming method (priming <65%), based on cut-off points used in previous research (*Keogh and Pearson, 2018*).

Here, we again found a strong effect of stimulus luminance in the perceptual phase of the task for the aphantasic participants (*Figure 3A*: perception section; $F(1, 17) = 81.18$, $p < 0.001$), reflecting a functional pupillary light response. However, we found no significant effect of luminance on pupil size during imagery *Figure 3A*, box insets: imagery section; $F(1, 17) = 0.193$, $p = 0.67$ and *Figure 3B* shows the lack of pupil diameter change for bright stimuli (red bars) and dark stimuli (blue bars). Similarly, to the general population, there was no main effect of set size during perception $F(1, 17) = 1.92$, $p = 0.18$, however interestingly, there was a significant main effect of set size during imagery $F(1, 17) = 6.185$, $p = 0.02$, with greater pupil diameters for Set-Size-Four compared to Set-Size-One (when averaged across the brightness conditions). This suggests that the aphantasic participants were

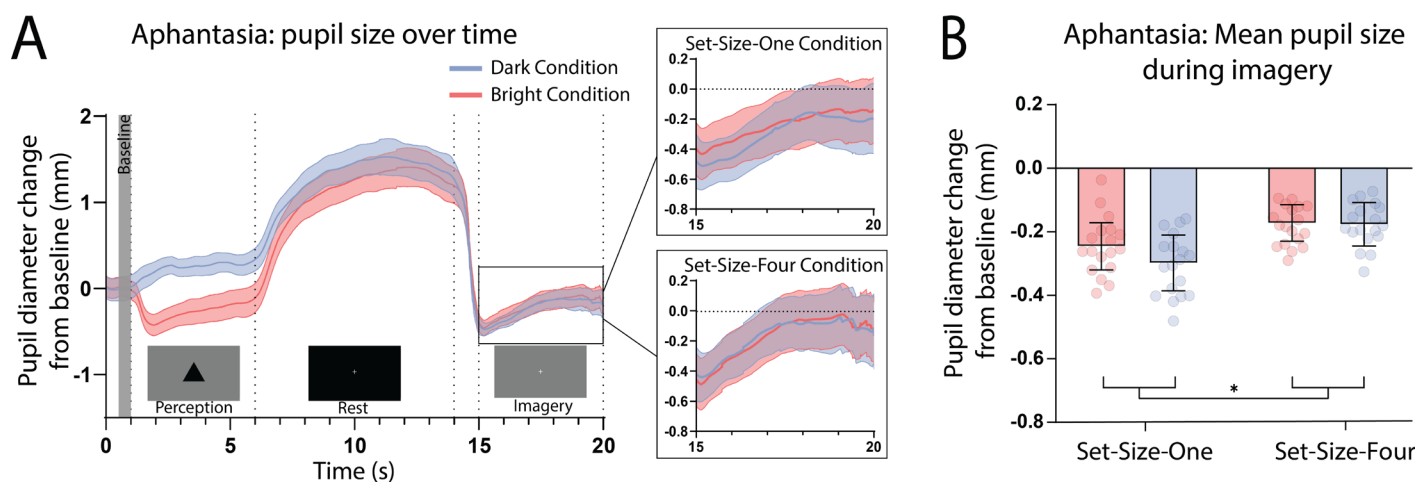

**Figure 3.** Pupillary response eye-tracker results for the aphantasic population. (**A**) Mean pupil size waveforms over time. *Left panel:* data averaged across the course of a trial for Bright (red lines) and Dark (blue lines) conditions for the aphantasic population. *Right panels:* Set-Size-One and Set-Size-Four conditions are shown separately during the imagery period. (**B**) Mean pupil size change from baseline during imagery (i.e. averaged from seconds 15 to 20 of trials) of Bright (red bars) and Dark stimuli (blue bars). Error bars indicate ± standard error of the mean (SEM), calculated across participants. *p < 0.05.

The online version of this article includes the following source data and figure supplement(s) for figure 3:

**Source data 1.** Source data for *Figure 3B*.

**Figure supplement 1.** Mean vividness ratings for set size and group.

**Figure supplement 2.** Pupil diameter changes during perception for control and aphantasic participant's.

**Figure supplement 3.** Eccentricity values for control and aphantasic individuals.

**Figure supplement 4.** The number of saccades during perception for control and aphantasic participant's.

**Figure supplement 5.** The number of saccades during imagery for control and aphantasic participant's.

actively engaging in the imagery task and exerting greater cognitive effort for the larger set size (*van der Wel and van Steenbergen, 2018*). In comparison to the general population, 61.11% (11/18) of the aphantasic individuals had difference scores that were lower than or equal to 0 for set size one as compared to 9.5% (4/42) of the general population (see *Figure 2B*). To confirm this absence of an imagery effect in the aphantasia population, we compared the pupil-difference score obtained when comparing the bright and dark conditions for the control and aphantasia groups, and computed a Bayes Factor (H0: score = 0; H1: score ≠ 0; see Materials and methods). Controls showed very strong evidence for H1 ($BF_{10} > 10^{10}$; Bayesian one-sample *t*-test), whereas the aphantasia population showed evidence for the null effect ($BF_{01} = 3.180$). A direct comparison between the control and aphantasia groups using a Bayesian repeated measure analysis of variance (ANOVA; see Materials and methods) showed very strong evidence for an effect of group ($BF_{10} > 10^6$). Finally, and as expected, pupil-difference scores (imagery of dark stimuli–bright stimuli) did not significantly predict imagery strength (measured using the binocular rivalry paradigm) for the aphantasic population (*Figure 2B*: yellow triangles; Set-Size-One: $r_p(17) = 0.20$, p = 0.44); Set-Size-Four: ($r_p(17) = -0.08$, p = 0.76). It should be noted that we could not perform an analysis on the vividness data in the same way as was done with the general population (*Figure 1D*) as the aphantasic individuals did not have any variation in their vividness ratings, reflecting their lack of subjective visual imagery (see *Figure 3—figure supplement 1*).

Age disparities between the groups are a potential confounding variable. This factor is of particular importance because the sensitivity of the pupillary light response, as well as maximum pupillary constriction velocity and acceleration, are thought to decline with age, beginning at 40–50 years old (*Fotiou et al., 2007*; *Lobato-Rincón et al., 2014*). However, trial time-course pupil waveforms are very similar for both general and aphantasic populations (*Figures 1B and 3A*, respectively). Both groups exhibited similar levels of pupil change during the perception phase of the task. Furthermore, a two-way ANCOVA was run on pupil-difference scores between general population and aphantasic

groups with age as a covariate. Levene's test and normality checks were carried out and the assumptions were met. We found a significant difference in pupil-difference score ($F$(1, 57) = 4.763, p = 0.033) between the groups when accounting for age. This provides evidence that decreased pupil responsiveness with age was not driving the observed effects.

Another possible explanation of our findings could be that the passive viewing of the perceptual images, lingering visual persistence and sluggish pupil responses could be driving our results. If this is the case, we would expect that pupil diameter during the perception of the images should correlate with pupil size during imagery for the corresponding images. Further, the pupillary light reflex during perception should be more pronounced in the control than the aphantasic populations. To investigate this possible alternative explanation of our data we first assessed the correlations between pupil diameter during perception of bright and dark images for Set-Size-One and -Four and their corresponding imagery conditions (control participants only). We found there were no significant correlations between any of the perception and imagery conditions, or the difference scores for set size one and four (all p > 0.40, see *Figure 2—figure supplement 1*). This lack of a correlation suggests that those individuals who have the largest pupillary light response while viewing the images, do not also have the greatest imagery driven pupillary light responses, making it unlikely that the pupil response while seeing the image is driving the mental imagery pupillary response. Next, we assessed whether the aphantasic individuals demonstrated any significant difference in their pupil responses to perceptual stimuli by running a 2 (image: bright and dark) × 2 (set size: 1 and 4) × 2 (group: aphantasic and controls) repeated measures ANOVA on the pupil diameter during the 5-s perceptual period of the task (see *Figure 1A* for task timeline). There was no main effect of imagery group $F$(1, 58) = 1.15, p = 0.29 and no significant interactions between imagery groups and any other factor (all p > 0.22, see *Figure 3—figure supplement 2*). These findings suggest the observed pupil responses during the imagery period of the task is unlikely to be a carry-over effect of the previous sensory response to perceived images.

Pupil size has been shown to depend on eye position (*Drewes et al., 2014*; *Gagl et al., 2011*) and the preparation to make a saccade to an upcoming image (*Jainta et al., 2011*; *Mathôt et al., 2015*; *Wang et al., 2018*). Pupil modulation and eye position are also both controlled by largely overlapping circuitry (*Wang and Munoz, 2018*). It could then be the case that group differences in eye position or saccades (either while viewing or imagining the triangles) may explain our data. To assess how eye movements and position (eccentricity) might be related to our findings, we analysed both eye position and saccades made while viewing and imagining the images, to see if these differed as a function of group. Eccentricity was extracted using the 'saccades' package in R (*von der Malsburg, 2015*) and saccades were detected using a velocity-based algorithm (*Engbert and Kliegl, 2003*) using the same R package. There were no significant differences between the groups for the number of saccades made during perception or imagery of the stimuli (see *Figure 3—figure supplement 4* and *Figure 3—figure supplement 5*). There were also no differences in mean eccentricity values when comparing the two groups (*Figure 3—figure supplement 3*), and no correlation with eccentricity and the pupillary light response (*Figure 2—figure supplement 3*), binocular priming (*Figure 2—figure supplement 4*), or vividness ratings (*Figure 2—figure supplement 5*), suggesting that differences in fixation or eye movements between the two groups is unlikely to drive the observed group differences in regard to the mental imagery pupillary light response.

Taken together these data from the general population and aphantasic individuals suggest that it is the content and ability to form vivid visual images, not the voluntary attempt to do so or the semantic content, that is driving the imaginary pupillary light response, providing the first evidence that these pupil changes are due to the sensory strength of imagery content and are not driven by higher-level semantic content.

## Discussion

Our results provide novel evidence that our pupils respond to the vividness and strength of a visual image being held in mind, the stronger and more vivid that image, the greater the pupillary light response. Our data provide the first evidence linking the pupil response to strength and vividness of imagery, not only between individuals, but also within an individual as imagery vividness fluctuates from moment to moment (*Dijkstra et al., 2017*; *Pearson et al., 2011*; *Rademaker and Pearson,*

*2012*). Finally, we show that, as a group, there is no evidence of this pupil response in individuals without mental imagery (aphantasia).

How might the content of mental imagery be driving the pupillary light response? One interpretation of these findings is that this imagery pupillary response is a by-product of the top-down modulation of midbrain-level visual circuitry (pretectal olivary nucleus, superior colliculus; *Joshi and Gold, 2020*), which occurs when imagining vividly, resulting in these regions interpreting this modulation as coming from external or afferent stimuli, and responding accordingly (*Larsen and Waters, 2018*; *Schwalm and Rosales Jubal, 2017*). In this case, the pupil would be responding to imagined luminance in much the same way that it responds to retina-based light sources. This is consistent with current data and models proposing shared mechanisms between visual imagery and perception (*Dijkstra et al., 2017*; *Dijkstra et al., 2019*; *Ganis et al., 2004*; *Naselaris et al., 2015*; *Xie et al., 2020*) and the idea that visual imagery functions much like a weak version of afferent perception (*Pearson, 2019*), supporting the idea that the stronger or more vivid an individual's imagery is, the more 'perception like' their imagery is.

An alternative mechanistic account might be that pupil diameter is encoded along with the original visual information for example bright object, and hence is replayed during memory decoding to form the mental image. This would be in a similar manner to theories proposing a functional role of eye movements during imagery generation from memory (*Wang et al., 2020*). It will be up to future work to uncover the exact mechanist account of imagery induced pupil changes.

Here we also provide the first objective physiological evidence of an extreme lack of visual imagery in aphantasic individuals. Aphantasia has largely been defined using subjective means (*Dawes et al., 2020*; *Jacobs et al., 2018*; *Pounder et al., 2018*; *Zeman et al., 2015*, but see *Keogh and Pearson, 2018*). Accordingly, people have remained sceptical about its true nature and possible psychogenic basis (*de Vito and Bartolomeo, 2016*). Our data demonstrate that using a non-visual strategy (no imagery in aphantasia) to think about bright and dark objects does not induce a pupillary light response. These data simultaneously provide strong evidence linking the pupillary light response to mental imagery, as well as supporting the behavioural work showing that aphantasic individuals indeed lack visual sensory imagery (*Keogh and Pearson, 2018*). Because the pupillary light response is involuntary (*Bouffard, 2019*), we can consider these findings as an unbiased neurophysiological measure of aphantasia. Not only do these data show that pupillary light response can be an objective index of imagery strength in studies of imagery in general populations, our data also provide a new low-cost objective measure for aphantasia that is uniquely based on a physiological mechanism and not reliant on self-report.

Could a lack of active engagement during imagery explain the aphantasia results? Put another way, are such participants refusing to imagine (*de Vito and Bartolomeo, 2016*)? We think this is highly unlikely as pupil size *did* increase as a function of set size for aphantasic individuals when attempting imagery, as has previously been shown in the general population, demonstrating the typical relationship between cognitive effort or arousal and pupil dilation (*Kahneman and Beatty, 1966*; *van der Wel and van Steenbergen, 2018*). This demonstrates active task engagement, suggesting that aphantasic individuals were not simply 'refusing' to actively participate in the task due to demand characteristics or a belief that they are unable to imagine (*de Vito and Bartolomeo, 2016*).

Further, we ran Bayesian one-sample *t*-tests on the binocular rivalry and pupillary light response difference scores (see *Figure 2*) comparing their performance to chance to see if there was any evidence they were performing significantly below chance. We found no significant evidence of below chance performance for either group on either the binocular rivalry or pupillometry imagery tasks (see *Figure 2—figure supplement 7*). Taken together, with the set-size pupillary effect we observed in our aphantasic participants, it seems unlikely that our aphantasic individuals were not engaging in the tasks. However, we cannot fully rule out this possibility. Further, there was no significant evidence of an abnormal pupillary response in our aphantasic cohort when viewing images, thus it is likely the lack of an imaginary pupillary light response is due to their lack of visual imagery. It also reveals that regardless of what imagery strategy aphantasic participants are implementing (e.g. propositional, spatial, language-like) to recall information about the shapes, they require greater cognitive effort to simultaneously maintain a larger number of shapes in their mind.

One limitation of our study is we did not include catch trails in our pupillometry task, that is we did not include trials where we asked participants to report on what image they had been asked to

imagine. We did however include catch trials in our binocular rivalry task through presenting mock binocular rivalry trials. If aphantasic participants are showing a response bias we would expect see a reduction in these mock priming trials when compared to the control population, which we did not find (see *Figure 2—figure supplement 6*). Adding catch trials to future experiments, in addition to set-size manipulations, may help to further confirm participant engagement. However, adding a simultaneous memory component to the task may lead some subjects to use a non-visual imagery strategy and as such, a reduction or dilution of the pupillary light response (see *Pearson and Keogh, 2019*). Future studies of visual imagery, and even more importantly when investigating aphantasia, should aim to include appropriate positive controls that allow for the identification of task engagement even when an individual doesn't have visual imagery. This will allow researchers to exclude the alternate explanation that those individuals who do not show evidence of imagery are not just refusing to imagine or not completing the task correctly.

Another possible explanation of our results is that perceptual pupillary light responses are lingering throughout each trial and driving the observed imagery pupil response. If this is the case, then pupil responses during perceptual viewing and imagery should be correlated, however we did not find any such correlations (see *Figure 2—figure supplement 1*). Further, when directly comparing the perceptual pupil responses between the general population and aphantasic individuals, there was no main effect of group or interaction between group and stimuli brightness or set size (see *Figure 3—figure supplement 2*). This demonstrates that there is no significant difference in the perceptual pupillary responses between the two groups, making it unlikely that aphantasic individual's lack of an imagery pupillary response is due to a lack of perceptual response. Finally, we also asked participants if they perceived any after images during the imagery period and any participants who reported they did were excluded from the study. Taken together, these results suggest that it is unlikely that the pupillary response to perceptually viewing the images is driving our observed imagery pupillary responses, and the lack thereof in the aphantasic individuals. Instead, it appears the pupillary light response during the visual imagery period reflects the wilful generation of imagery in the mind's eye of those who experience visual imagery. This is further substantiated by the strength of visual imagery (measured using the binocular rivalry paradigm) correlating with the imagery pupillary light reflex, but not the perceptual pupillary light reflex (see *Figure 2B* and *Figure 2—figure supplement 2*).

We also found that in the imagery task, higher within-trial reports of vividness are reflected by greater pupillary light responses (within-subjects effects; see *Figure 1D*). This indicates that participants were able to accurately evaluate the vividness of individual episodes of imagery in comparison to other vividness episodes on previous trials. However, average vividness ratings did not correlate with their pupil-difference scores, that is, participants who gave higher vividness ratings on average did not necessarily have increased pupil light responses in response to imagery (between-subjects effects; see *Figure 1—figure supplement 2*). Participant's scores on the VVIQ also did not correlate with pupil-difference scores (between-subjects effects; see *Figure 1—figure supplement 3*). This suggests that participants might have difficulties in accurately reporting their strength of sensory visual imagery on an absolute scale (i.e. from 'no image' to 'as vivid as perception'), and brings into question the reliability of these subjective measures of imagery and highlights the utility of using objective or online (i.e. in a task), and less trait-like measures when studying visual imagery.

Recent studies have shown pupil size is also modulated by the content of visual working memory (*Blom et al., 2016*; *Hustá et al., 2019*; *Zokaei et al., 2019*). It is interesting to note here that previous work has shown that imagery has been implicated as one mnemonic that can be used to retain information in mind during visual working memory tasks (*Albers et al., 2013*; *Keogh and Pearson, 2011*; *Keogh and Pearson, 2014*; *Keogh and Pearson, 2017*). This highlights the possibility that it is imagery, being used as a mnemonic strategy, that is driving the pupillary light response observed in visual working memory experiments (*Pearson and Keogh, 2019*). Although many participants report using a visual imagery strategy during these tasks, some participants report using a non-visual imagery strategy when remembering visual information, and recent work demonstrates that aphantasic individuals can perform traditional visual working memory tasks just as well as control populations (*Keogh et al., 2021b*). Measuring the pupillary light response in aphantasic individuals, and those who report not using an imagery strategy, while performing classic visual working memory tasks may help to further elucidate these differences in cognitive strategy use in a more objective manner.

One limitation that is important to note here is that our aphantasic sample contained a relatively small sample (18 participants) due to the relative rarity of this condition. Further our two samples were not age matched, which may have affected our results, however seeing as there was no difference between the two groups for the perceptual pupillary light response, we think this is unlikely to be driving our findings. Future studies should aim to replicate and extend these findings with a larger group of aphantasic individuals and age matched controls.

To conclude, the present study demonstrates that the pupillary light response can be used as a physiological index of individual differences in the sensory and phenomenological strength of visual imagery, including the lack of visual imagery – aphantasia. Combining this measure with the binocular rivalry paradigm in favour of subjective alternatives will increase the reliability and objectivity of imagery test batteries and may lead to the development of more congenial theories of the mind's eye.

## Materials and methods

### Participants

Fifty-six psychology students with a mean age of 19.8 years (range 18–31, 27 females) were recruited for the study and participated for course credit. We aimed to obtain analysable data from a minimum of 40 participants, which should be a large enough sample to identify a strong positive correlation between pupil dilation and imagery, which is what we would expect if imagery content were driving the previously observed imagery pupillary light response (g*Power effect size = 0.5, $\alpha$ = 0.05, $\beta$ = 0.95). Fourteen of these participants were excluded from data analysis for not meeting a priori criteria (see Exclusion criteria), leaving 42 participants in the final general population sample.

The aphantasic individuals come from a rare population and for this reason we did not run a specific power analysis but aimed to collect a minimum of 15 participants. We had nineteen aphantasic individuals agree to participate in the study with a mean age of 35.8 years (aged 18–54, 12 females). One of these individuals was excluded from data analysis for not meeting a priori criteria (see Exclusion criteria), leaving 18 in the final sample. These participants had all contacted the lab regarding their aphantasia and asked to participate in our research. They were all reimbursed $20 AUD per hour for their participation. All participants had normal or corrected to normal vision (i.e. glasses or contacts). Both experiments were approved by the UNSW Human Research Ethics Advisory Panel (HREAP-C 3182).

### Apparatus

Apparatus stimuli in all experiments were presented on an LCD display monitor (Dell UltraSharp U2419H) with 60 Hz refresh rate and a 1920 × 1080 resolution. Luminance values of all stimuli were measured using a Konica Minolta chroma meter (CS-100A). Participants placed their chin on a chin rest throughout the experiment to maintain fixation at a distance of 57 cm from the monitor 13 and to limit head movements. The tasks were performed in a blackened room to eliminate any possible fluctuations in ambient light.

In the **pupillary response task**, pupil sizes and eye movements were recorded using head mounted eye-tracking glasses (Pupil, Pupil Labs GmbH, Berlin, Germany) (*Kassner et al., 2014*). Pupil diameter of participants' right eye was continuously sampled at 200 Hz throughout the task. A pupil detection 3D algorithm locates the dark pupil in the infrared illuminated eye camera image, thus recording capabilities are not compromised by an absence of room lighting. Pupil diameter is then scaled to millimetres (mm) based on mean anthropomorphic eyeball diameter and corrected for perspective. The algorithm does not depend on corneal reflection, and is compatible with users who wear contact lenses and most eyeglasses (*Kassner et al., 2014*).

A second camera mounted on the glasses continuously recorded participants' field of view. Footage from this camera was subsequently assessed to ensure fixation on the computer monitor was maintained throughout the task. The experiment was designed using MATLAB (version R2017b). ZeroMQ plug-ins were used for cross-communication between eye-tracking and stimulus presentation platforms (*Akgul, 2013*). Pupil data were recorded with Pupil Capture v.1.10.20 (Pupil Labs) installed on an ASUS (GL502V) PC (Windows 10).

In the **binocular rivalry task**, participants wore red-green anaglyph glasses to ensure rivalrous stimuli were presented to left and right eyes in isolation. Responses of 1, 2, or 3 on a keyboard were

used by participants to indicate which image dominated their perception during binocular rivalry (1 for green; 3 for red; 2 for perceptually mixed green and red).

## Stimuli

For the **pupillary response task**, 32 achromatic shape stimuli were created for participants to perceive and then later imagine in their absence, across 32 trials. The stimuli were evenly divided based on a 2 × 2 factorial design, belonging to one of two luminance conditions ('Bright' or 'Dark') and one of two set-size conditions ('Set-Size-One' or 'Set-Size-Four'). Shapes belonging to the Bright condition were either white with a luminance of 117 cd/m² or light grey with a luminance of 65 cd/m². Shapes in the Dark condition were black (1 cd/m²) of dark grey (9 cd/m²). Set-Size-One stimuli consisted of a single equilateral triangle with 12.5 cm sides, subtending 12.5° of visual angle. Set-Size-Four stimuli consisted of an arrangement of four smaller equilateral triangles with a total surface area and luminance equal to that of the corresponding Set-Size-One triangles (see *Figure 1—figure supplement 1* for illustration of all stimuli). Stimuli were also uniquely orientated at either 0°, 90°, 180°, or 270° (e.g. four Set-Size-One black triangles, each with a different orientation. See *Figure 1—figure supplement 1* for examples all possible shape orientations). This ensured that all 32 stimuli were unique and participants and were encoding information about a new stimulus on each trial, therefore avoiding the use of long-term memory. Set-Size-Four stimuli therefore subtended either 10.8° or 18.9° of visual angle depending on their orientation.

All stimuli were presented on a grey background screen with a luminance of 26 cd/m². This same level of background luminance was used during measurement of baseline and imagery phases. A fixation cross on a black background with a luminance of 1 cd/m² was presented during the resting phase of each trial. All stimuli were created in MATLAB, using the Psychophysics Toolbox 3 extensions (*Brainard, 1997*).

For the **binocular rivalry task**, sinusoidal luminance modulated Gabor patterns were used as rivalrous stimuli; vertical-green (CIE chromaticity coordinates: $x = 0.275$, $y = 0.590$) and horizontal-red (CIE chromaticity coordinates: $x = 0.492$, $y = 0.372$), both with a mean luminance of 8.35 cd/m² and 7.1° of visual angle. In each trial, both patterns were presented at the same time around a fixation point at the centre of a black background screen. Mock rivalry stimuli (a single 15 Gabor pattern spatially divided into half vertical-green and half horizontal-red) were used on 12.5% of trials to measure the influence of decisional bias or lack of attention to the task. More details on the binocular rivalry task can be found in *Keogh and Pearson, 2014*.

All participants also complete the VVIQ (*Marks, 1973*) to get a self-report measure of trait imagery vividness (see *Figure 2—figure supplement 8* for VVIQ data for general population and aphantasic individuals).

## Procedure

### Pupillometry imagery experiment timeline

Each trial began with the presentation of a white fixation cross at the centre of a grey screen (baseline) for 1 s. An image was then presented at the centre of this grey screen for 5 s (either one or four triangles of varying brightness, see *Figure 1—figure supplement 1* for illustrations of all stimuli). Participants were instructed to focus on the stimuli during this time and memorize its size, orientation, and level of brightness. Next, a black screen with a white fixation cross was presented for 8 s, allowing the perceived after-image to completely fade and pupils to dilate back to equivalent resting levels. The grey baseline screen was then presented again for 6 s. During this time, participants were cued (via two auditory beeps) to actively imagine the stimuli observed previously during that trial. Lastly, participants were prompted to report the vividness of their imagery during those previous 5 s on a scale of 1–4 (1 being 'not vivid at all – no shape appeared in imagery'; 4 being 'very vivid – almost like seeing it') via key response.

### Binocular rivalry paradigm

Participants were cued to imagine either a red or green Gabor pattern prior to binocular rivalry with the letter 'R' or 'G' (750 ms). Participants then imagined the image for 6 s, after which they were presented with the binocular rivalry display (750 ms) and were asked to indicate which image was

dominant (see *Figure 2*). Trials where participants reported seeing the pattern they were cued to imagine as dominant were denoted as 'primed' trials. Participants completed 2 blocks of 48 trials resulting in a total of 96 trials in total (84 real and 12 mock trials). The number of primed trials divided by the total number of trials (excluding mock trials and mixed percepts) was used to calculate a percent primed score for each participant. Mock trial priming was calcuated by giving a value to each mock trial as either 0 (reporting the catch trial as the opposite colour to that primed), 50 (reporting the catch trial as being mixed), or 100% (reporting the catch trial to be the same as the cued image) (*Figure 2—figure supplement 6*). These values are then averaged to get a priming value where 50% indicates no bias, while higher values indicate a bais towards reporting the mock trails as being the same as the imagined image, while negative numbers indicate a bias towards reporting the oppoiste image to that which was imagined.

## Exclusion criteria

Of the 56 participants recruited for the general population sample, 14 in total were excluded from data analysis due to not meeting a priori criteria.

Pupillary response task exclusions: eight participants were excluded because more than 50% of their pupil data points were below the pupil detection algorithm confidence value of 0.6, provided by the Pupil Capture system. This cut-off point was derived prior to data collection and is the recommended cut-off point for obtaining accurate pupil size data (Pupil Labs). Three participants were excluded due to reporting (during systematic post-task questioning) seeing after-images of the shape stimuli for longer than the 8 s black screen presentation (i.e. seeing after-images during the imagery phase of trials), because pupil size is known to be influenced by the induced compensatory light perception of an after-image (*Tsujimura et al., 2003*).

Binocular rivalry task exclusions: three participants were excluded due to having mock rivalry priming >66.67% (more than one incorrect response on the mock trials), which indicated either an influence of decisional bias or lack of attention to the task. An a priori cuff-off point of scoring *both* below 65% priming on the binocular rivalry task and below 32 on the VVIQ was used to exclude participants who potentially did not have visual imagery (i.e. may be aphantasic). No participants fell below this combined cut-off point thus none were excluded on this basis.

Of the 19 participants recruited for the aphantasic population, 1 was excluded from data analysis because more than 50% of their pupil data points were below the pupil detection algorithm confidence value of 0.6, given by the Pupil Capture system. All participants scored below both of the a priori cut-off points of 32 on the VVIQ and 65% on the binocular rivalry task, therefore, no participants were excluded due to this criterion.

## Data analysis

For the pupillary response task, cubic spline interpolation was used to estimate pupil diameter during periods where subjects' pupils were occluded due to blinking (in accordance with *Mathôt et al., 2013*). Artefacts in the pupil data were then smoothed using a moving average Hanning window (*Kret and Sjak-Shie, 2019*). Individual trials in which mean pupil diameter while passively viewing the grey baseline screen was lower than 2 mm or higher than 8 mm were excluded (*N*(total trials from whole sample) = 8) as values outside this range are unnatural pupil sizes and were clear outliers based on inspection of participants' pupil-baseline histograms (*Mathôt et al., 2018*). Trials were averaged to form condition-specific pupil diameter waveforms to represent change in pupil size over time. Mean pupil diameter values during imagery in each trial were baseline corrected using a within-trial baseline subtraction approach (*Mathôt et al., 2018*) (i.e. subtracted from mean pupil diameter during 0.5 s prior to stimulus perception onset) to account for temporal shifts in pupil size across the experimental session due to fatigue (*Morad et al., 2000*). A two-way repeated measures ANOVA was used to compare Dark and Bright means during perception and imagery within both set-size conditions. 'Pupil-difference' scores were calculated by subtracting Dark condition means from Bright condition means of the corresponding set size for comparison with binocular rivalry percent primed scores. Pupil-difference scores were also separated based on the discrete within-trial vividness ratings to assess metacognition and whether pupil size changes in response to imagery were reflective of subjects' own experience of vividness of visual imagery.

In the binocular rivalry task, trials where participants reported seeing the pattern they were cued to imagine as dominant in the subsequent binocular rivalry display were denoted as 'primed' trials. The number of primed trials divided by the total number of trials (excluding mock trials and mixed percepts) was used to calculate a percent primed score for each participant. Participants' percent primed scores in binocular rivalry were correlated with their pupil-difference scores (both Set-Size-One and Set-Size-Four) to assess potential for the pupillary response task to measure individual variability in visual imagery strength.

The LMEs were run in R (*R Development Core Team, 2018*) using the lme4 package and ANOVA's and ANCOVA were run in SPSS v.25.0 (*IBM Corp. Released, 2017*). For the linear mixed-effects models set size (1 or 4) and vividness ratings (1, 2, 3, and 4) were entered into the model as fixed effects. As random effects intercepts for subjects were entered into the model. p values were obtained by likelihood ratio tests of the full model with vividness included vs. the model without vividness included.

Bayesian statistics were used to determine whether null findings can be interpreted as evidence for an absence of effect (*Dienes, 2014*). We used Bayesian repeated measure ANOVA (within-subject effect: set size; between-subject effect: group) to compare the control and aphantasia groups as well as Bayesian one-sample *t*-tests to compare each group with H0, defined as the absence of effect. All Bayesian analysed were performed with JASP (Version 0.10.2).

## Additional information

### Funding

| Funder | Grant reference number | Author |
| --- | --- | --- |
| National Health and Medical Research Council | APP1024800 | Joel Pearson |
| National Health and Medical Research Council | APP1046198 | Joel Pearson |
| National Health and Medical Research Council | APP1085404 | Joel Pearson |
| National Health and Medical Research Council | APP1049596 | Joel Pearson |
| Australian Research Council | DP140101560 | Joel Pearson |
| Human Frontier Science Program | LT000362/2018-L | Thomas Andrillon |

The funders had no role in study design, data collection, and interpretation, or the decision to submit the work for publication.

### Author contributions

Lachlan Kay, Conceptualization, Data curation, Formal analysis, Investigation, Methodology, Visualization, Writing – original draft; Rebecca Keogh, Conceptualization, Data curation, Formal analysis, Investigation, Methodology, Project administration, Supervision, Visualization, Writing - review and editing; Thomas Andrillon, Methodology, Software, Supervision, Writing - review and editing; Joel Pearson, Conceptualization, Funding acquisition, Methodology, Project administration, Resources, Supervision, Writing - review and editing

### Author ORCIDs

Rebecca Keogh (ID) http://orcid.org/0000-0003-4814-433X
Thomas Andrillon (ID) http://orcid.org/0000-0003-2794-8494
Joel Pearson (ID) http://orcid.org/0000-0003-3704-5037

### Ethics

Informed written consent was obtained from all participants to participate in the experiment and to publish their anonymized data in a journal article. Both experiments were approved by the UNSW Human Research Ethics Advisory Panel (HREAP-C 3182).

### Decision letter and Author response

Decision letter https://doi.org/10.7554/eLife.72484.sa1
Author response https://doi.org/10.7554/eLife.72484.sa2

## Additional files

### Supplementary files

• Supplementary file 1. Fixed effects estimates for pupil-difference scores as a function of vividness ratings. This file provides the fixed effects estimates for the LME run on the pupil-difference scores for the general population as a function of vividness ratings and set size.

• Transparent reporting form

### Data availability

Figure 1 - Source Data 1& 2, Figure 2 - Source Data 3, and Figure 3 - Source Data 4 contain the numerical data used to generate the figures.

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
