## [Editor Report]

This is a rigorous study of the relation between the vividness of visual imagery and the pupillary light response that can result from it. It provides evidence for the absence of imagery in individuals that self-report as aphantasic. The results will likely be of interest to researchers in a range of disciplines such as psychology, neuroscience and philosophy.

---

## [Decision Letter]

**Decision letter after peer review:**

Thank you for submitting your article "The eyes have it: The pupillary light response as a physiological index of aphantasia, sensory and phenomenological imagery strength" for consideration by *eLife*. Your article has been reviewed by 2 peer reviewers, and the evaluation has been overseen by a Reviewing Editor and Chris Baker as the Senior Editor. The following individuals involved in review of your submission have agreed to reveal their identity: Martin Rolfs (Reviewer #2); Jessica Breedlove (Reviewer #3).

We all thought that the paper was interesting and that it addresses a timely topic with a novel approach. There is a lot to like here in terms of learning about mental imagery but also about people who seem to lack it. That said, you'll see that the reviewers brought up some substantive points, and, after consultation, we think that a revision might be possible to address these points but that it will require more analyses and most likely more data.

In consultation, we focused primarily on two issues: eye movements and demand characteristics.

With respect to eye movements, there are a number of reasons why we think further analyses are crucial:

1. Pupil size (in particular, constriction velocity, maximum constriction, and mean pupil change) depends stimulus eccentricity. (e.g., https://doi.org/10.1016/j.visres.2020.03.008)

2. Measured pupil size depends on eye position, and vice versa. The origin of these effects are measurement errors related to video-based eye tracking, as it this dependency is seen even for artificial, fixed-size pupils (e.g., https://doi.org/10.3758/s13428-011-0109-5; https://doi.org/10.1371/journal.pone.0111197).

3. There is now plenty of evidence that saccade preparation alters pupil size (https://doi.org/10.1037/a0038653; https://doi.org/10.3389/fnhum.2011.00097; https://doi.org/10.5334/joc.33; https://doi.org/10.1111/ejn.12883) and that both are controlled by largely-overlapping circuitry (e.g., https://doi.org/10.1073/pnas.1809668115).

4. Saccades greatly alter content in visual short-term memory (e.g., https://doi.org/10.1037/xlm0000338; reviewed in https://doi.org/10.1080/13506285.2020.1764156).

5. Eye movements can be correlated with imagery (e.g., https://doi.org/10.1162/jocn.1997.9.1.27); comparison between groups might thus provide additional indicators of imagery.

As a consequence, a comparison of eye position and saccade statistics is very important.

With respect to demand characteristics, R3 has some detailed suggestions, and we all feel that a revision of the design would potentially yield significant gains. It doesn't seem technically difficult to collect the data, but we also understand that collecting any data in person these days can be challenging. However, at the very least we think you should report on the mock trials in the binocular rivalry task (to show there wasn't a difference between groups in what you did check for, a BR bias), pull back on your claims accordingly, and provide a discussion of this limitation to the results.

Again, please see the reviews for more details, but we hope that you will find the comments helpful in deciding next steps for the paper.

*Reviewer #1 (Recommendations for the authors):*

1. Please analyze and report eye movement parameters in each experiment and whether correlate with vividness of imagery. Please also clarify where observers were required to look during each phase of the experiments.

2. Please provide evidence that the correlations reported in Figure 1D persist at the level of individuals.

3. Please revise this Discussion section to clarify that such tests would always have to be combined with positive tests that show the commitment of participants to the task instructions.

*Reviewer #2 (Recommendations for the authors):*

My largest (and maybe only) major concern is the possibility that aphantasic subjects were not attempting to imagine during both the main experiment and the BR task. This concern is somewhat exacerbated by the fact that subjects reached out to the lab to participate, making risk for demand characteristics high. The increase in pupil size in the aphantasic group in the set-size-4 condition is encouraging but this increase could also represent a number of other things. While I am hard pressed to come up with a way to ever eliminate this potential fully, I do think a bit more could be done to strengthen the argument against lack of participation and so I'm wary of the claim that this has been fully "ruled out" (line 322). My suggestions are as follows:

– Most ideally, there would be an addition of a probe task following the imagery period that would require subjects to report on the objects that they were supposed to remember and imagine (for example, like the one the in Zokaei 2019 which the authors cited). Both groups should be able to perform fairly well since, as the authors point out, aphantasic individuals can perform visual working memory tasks just as well as the general population. This would serve as a better indicator of effort and attention and require that subjects at the very least engage with the visual features of the stimuli (maybe they would have to adjust a probe's brightness until it matched the imagined one or adjust an angle to match that of the triangle, etc.).

– I also appreciate the use of a more objective measure of imagery vividness with the binocular rivalry task, but this suffers from the same issue. Aphantasic subjects could be not attempting to imagine or have an unintentional bias pushing them away from indicating the priming effect. The latter can be addressed using catch/mock trials as the authors did use. However if my math is correct, there were 24 trials total with only 3 mock trials. So while they did exclude subjects for making more than 1 incorrect response to mock trials, that still means that subjects could have shown a bias on 33% of the trials and still be included in the study. This might account for some of the findings if there were more aphantasic than control subjects who incorrectly answered a mock trial. My suggestion is then to increase the number and type of mock trials (all green and all red in addition to mixed) and (or at the very least) report the missed mock trials for each group.

– I would also suggest estimating the null hypothesis and significance bound through random sampling of the experiment under the null to show that the aphantasia subjects did not perform significantly worse than chance in Figure 2b.

Lines 124 – 127 "These data provide novel evidence individuals can reliably evaluate the comparative vividness of single episodes of imagery. Further, these data demonstrate that the pupillary light response also tracks the phenomenological vividness of visual imagery from moment to moment." These two lines are circular. The authors are using changes in pupil size as evidence that subjects can reliably report their vividness and then using subject reports as evidence that pupil size is a good indicator of vividness. Suggest rewriting /rewording

Around line 256: the authors state there is no correlation between perception and imagery conditions. I initially found this confusing because if imagery is like faint vision, the pupil change during imagery of an object should mimic the pupil change during viewing of the same object. The point I believe they were making is that there is no correlation when all subjects are pooled together. I think this would be clearer if the authors first pointed out that there is a correlation between perception and imagery, but in the general population only (if true).

Line 312 – I believe the authors might be conflating psychogenic and fabrication and suggest that they revise the discussion to be clearer on this. The results speak to the latter but not so much the former. While psychogenic implies there might not be a traceable physical cause, it doesn't necessitate that the patient is acting intentionally and they likely experience it as real (for example, in non-epileptic seizures the brain is in fact not seizing, but the patient often isn't consciously convulsing and truly believes they are having a seizure). The lack of experience of imagery could lead to a lack of pupillary light response. A psychogenic source would not be inconsistent with the findings since the authors are not claiming to identify what is blocking or restricting the visual experience of imagery, just that there is a link between that experience and a physiological response. I suggest removing the claim that the study rules out a psychogenic source to aphantasia.

The timing diagram in Figure 1A seems off. The caption and methods say that the black screen rest was presented for 8s, but only 7s exist between the 2nd and 3rd dotted lines, and this doesn't match Figure 3. The reason for using a black screen for rest was also not clear. Together, these made the large dip in pupil size before imagery a bit confusing at first.

---

## [Author Response]

We all thought that the paper was interesting and that it addresses a timely topic with a novel approach. There is a lot to like here in terms of learning about mental imagery but also about people who seem to lack it. That said, you'll see that the reviewers brought up some substantive points, and, after consultation, we think that a revision might be possible to address these points but that it will require more analyses and most likely more data.In consultation, we focused primarily on two issues: eye movements and demand characteristics.With respect to eye movements, there are a number of reasons why we think further analyses are crucial:1. Pupil size (in particular, constriction velocity, maximum constriction, and mean pupil change) depends stimulus eccentricity. (e.g., https://doi.org/10.1016/j.visres.2020.03.008)2. Measured pupil size depends on eye position, and vice versa. The origin of these effects are measurement errors related to video-based eye tracking, as it this dependency is seen even for artificial, fixed-size pupils (e.g., https://doi.org/10.3758/s13428-011-0109-5; https://doi.org/10.1371/journal.pone.0111197).3. There is now plenty of evidence that saccade preparation alters pupil size (https://doi.org/10.1037/a0038653; https://doi.org/10.3389/fnhum.2011.00097; https://doi.org/10.5334/joc.33; https://doi.org/10.1111/ejn.12883) and that both are controlled by largely-overlapping circuitry (e.g., https://doi.org/10.1073/pnas.1809668115).4. Saccades greatly alter content in visual short-term memory (e.g., https://doi.org/10.1037/xlm0000338; reviewed in https://doi.org/10.1080/13506285.2020.1764156).5. Eye movements can be correlated with imagery (e.g., https://doi.org/10.1162/jocn.1997.9.1.27); comparison between groups might thus provide additional indicators of imagery.As a consequence, a comparison of eye position and saccade statistics is very important.

We thank the reviewers and editor for their detailed and thoughtful points regarding eye movements and pupil diameter. We think the points raised are fair and we have added an extra supplementary analysis to the manuscript analysing the eccentricity and saccade data. Assessing the eccentricity data we found that in general the participants mostly maintained fixation throughout the experiment, and there was no significant difference between the groups in their average eccentricity values (see Figure 2 —figure supplement 2). There was also no significant correlations between eccentricity and the pupillary light response for either group during the imagery period (see Figure 3 —figure supplement 3). We also assessed whether the number of saccades during perception and imagery was different across groups or as a function of either set size or luminance. We found that there were no consistent differences in the number of saccades across the two groups for these variables (see Figure 2 —figure supplements 3 and 4). We believe taken together these results suggest that it is unlikely that our pupil diameter findings are driven by different eccentricity/fixation or saccades between the two groups and we thank the Reviewers for helping us address this important potential alternative explanation of our data. We believe that by showing eye-movements are unlikely to be driving the observed imaginary pupillary light reflex our paper has been strengthened.

With respect to demand characteristics, R3 has some detailed suggestions, and we all feel that a revision of the design would potentially yield significant gains. It doesn't seem technically difficult to collect the data, but we also understand that collecting any data in person these days can be challenging. However, at the very least we think you should report on the mock trials in the binocular rivalry task (to show there wasn't a difference between groups in what you did check for, a BR bias),

Thank you for these suggestions. We will endeavour to incorporate their excellent suggestions into future studies we run. Due to the current testing environment however it will take a long time to collect enough data to run a full new study as suggested. As suggested, we have added in the mock trial data to the supplementary material (Figure 3 —figure supplement 5) and additional information regarding these trials (see response to point 3 from Reviewer #3).

pull back on your claims accordingly, and provide a discussion of this limitation to the results.

We have toned down our claims and added in discussion of the limitations of the study, see the response to point 2 from Review #3.

Reviewer #1 (Recommendations for the authors):1. Please analyze and report eye movement parameters in each experiment and whether correlate with vividness of imagery. Please also clarify where observers were required to look during each phase of the experiments.

Participants were instructed to maintain fixation on the fixation cross, throughout the experiment, which has been clarified in Figure 1’s legend. However, following Reviewer 2’s suggestion, we have now added the analysis of eccentricity and saccade data to the supplementary materials (Figure 2 —figure supplement 2 and 3, Figure 3 —figure supplement 3, see also above response to general points). We found that there were no consistent differences between the groups making it unlikely that differences in eyeposition is driving the lack of a pupillary light response in the aphantasic population.

2. Please provide evidence that the correlations reported in Figure 1D persist at the level of individuals.

We are not sure we understand the Reviewer’s comment correctly. We do not report correlations for Figure 1D, but the results of 2 x 4 linear mixed-effects analysis. This model included subject identity as a random effect (see Methods) and therefore the effects reported were computed at the subject level. We report in the text, effects that are significant at the level of the sample. This does not exclude the possibility of inter-individual differences, but we are not sure how interpretable a single-subject analysis is in the current study.

3. Please revise this Discussion section to clarify that such tests would always have to be combined with positive tests that show the commitment of participants to the task instructions.

We have now added the following to the discussion regarding the importance of including these commitment controls to imagery studies:

“Future studies of visual imagery, and even more importantly when investigating aphantasia, should aim to include appropriate controls that allow for the identification of task engagement even when an individual doesn’t have visual imagery. This will allow researchers to exclude the alternate explanation that those individuals who do not show evidence of imagery are not just refusing to imagine or not completing the task correctly.”

Reviewer #2 (Recommendations for the authors):My largest (and maybe only) major concern is the possibility that aphantasic subjects were not attempting to imagine during both the main experiment and the BR task. This concern is somewhat exacerbated by the fact that subjects reached out to the lab to participate, making risk for demand characteristics high. The increase in pupil size in the aphantasic group in the set-size-4 condition is encouraging but this increase could also represent a number of other things. While I am hard pressed to come up with a way to ever eliminate this potential fully, I do think a bit more could be done to strengthen the argument against lack of participation and so I'm wary of the claim that this has been fully "ruled out" (line 322).

We have re-worded parts of the discussion to tone down such claims:

“This demonstrates active task engagement suggesting that aphantasic individuals were most likely not simply ‘refusing’ to actively participate in the task due to demand characteristics or a belief that they are unable to imagine (de Vito and Bartolomeo, 2016)."

“…However, we cannot fully rule out this possibility. Further, there was no evidence of an abnormal pupillary response in our aphantasic cohort when viewing images, thus it is likely the lack of an imaginary pupillary light response is due to their self-reported lack of visual imagery.”

My suggestions are as follows:– Most ideally, there would be an addition of a probe task following the imagery period that would require subjects to report on the objects that they were supposed to remember and imagine (for example, like the one the in Zokaei 2019 which the authors cited). Both groups should be able to perform fairly well since, as the authors point out, aphantasic individuals can perform visual working memory tasks just as well as the general population. This would serve as a better indicator of effort and attention and require that subjects at the very least engage with the visual features of the stimuli (maybe they would have to adjust a probe's brightness until it matched the imagined one or adjust an angle to match that of the triangle, etc.).

Thank you for this excellent suggestion, which could be implemented in follow-up studies. However, as mentioned above, the current situation makes the planning of new experiments extremely uncertain. In addition, we did not find evidence suggesting aphantasic participants did not engage in the task. In fact, the modulation of pupil size by stimulus complexity suggests that these individuals engaged in the task, at least sufficiently for this effect to emerge. We agree that the methodology can be improved, and we are thankful for the Reviewer’s suggestion. But we do think our conclusions are warranted by the data at hand.

We have now further clarified our reasoning and outlined better the limitations of our study in the Discussion section. Indeed, we wanted the participants to focus more on holding the image in their mind and creating the most vivid image they were able to. Having them rate their vividness reinforces the imagery component of the task. If we had asked participants to remember the items instead, it is possible that some participants may have imagined the images as a mnemonic strategy. However, it is also possible that they may have also changed the type of strategy they used to remember the items which might not have involved imagery. This, in of itself, is interesting, however it was outside of the scope of this current study. In addition, if multiple participants were not usuing visual imagery to remember the images, this may have diluted the imaginary pupillary light response and replicating and extending this finding was central to the research question of this study. We have added in a limitations and future directions section to our discussion that speaks to these points:

“One limitation to our study is we did not include catch trails in our pupillometry task, i.e. we did not include trials where we asked participants to report on what image they had been asked to imagine. We did however include catch trials in our binocular rivalry task through presenting mock binocular rivalry trails. If aphantasic participants are showing a response bias we would expect see a reduction in these mock priming trails when compared to the control population, which we did not find (see Figure 3 —figure supplement 5). Adding catch trials to future experiments, in addition to setsize manipulations, may help to further confirm participant engagement. However, adding a simultaneous memory component to the task may lead some subjects to use a non-visual imagery strategy and as such, a reduction or dilution of the pupillary light response (see Pearson and Keogh (2019)). Future studies of visual imagery, and even more importantly when investigating aphantasia, should aim to include appropriate positive controls that allow for the identification of task engagement even when an individual doesn’t have visual imagery. This will allow researchers to exclude the alternate explanation that those individuals who do not show evidence of imagery are not just refusing to imagine or not completing the task correctly.”

– I also appreciate the use of a more objective measure of imagery vividness with the binocular rivalry task, but this suffers from the same issue. Aphantasic subjects could be not attempting to imagine or have an unintentional bias pushing them away from indicating the priming effect. The latter can be addressed using catch/mock trials as the authors did use. However if my math is correct, there were 24 trials total with only 3 mock trials. So while they did exclude subjects for making more than 1 incorrect response to mock trials, that still means that subjects could have shown a bias on 33% of the trials and still be included in the study. This might account for some of the findings if there were more aphantasic than control subjects who incorrectly answered a mock trial. My suggestion is then to increase the number and type of mock trials (all green and all red in addition to mixed) and (or at the very least) report the missed mock trials for each group.

We take the Reviewer’s point seriously. However, we note it is standard for mock or catch trials to represent a minority of the total number of trials. In fact, too frequent mock trials could make the subjects aware of the existence of the mock trials and fundamentally alter the results. In addition, we think our interpretation of the data (that the absence of priming is due to a lack of a reported imagery, in accordance with individuals’ self-reports) is more parsimonious than hypothesising the existence of an unintentional bias.

To give readers the clearest account of our data, we have now included the data for mock trials for both the controls and undergraduate students in the supplementary material in addition to VVIQ scores (Figure 3 —figure supplement 5 and Figure 2 —figure supplement 6). We have also further clarified how mock trials were calculated in the procedure as, upon re-reading the manuscript, we realised we did not include the number of trials participants completed in the binocular rivalry task, this has now been updated in the procedure section (84 real and 12 mock trials). The mock trials we use have a bespoke zig-zag walk border between the red and green patterns, and thus are not exactly the same each presentation, and appear slightly more red or green. We hope this explanation helps to clarify the mock trial data.

“Participants completed 2 blocks of 48 trials resulting in a total of 96 trials in total (84 real and 12 mock trials). The number of primed trials divided by the total number of trials (excluding mock trials and mixed percepts) was used to calculate a percent primed score for each participant. Mock trial priming was calcuated by giving a value to each mock trial as either 0 (reporting the catch trial as the opposite colour to that primed) , 50 (reporting the catch trial as being mixed) or 100% (reporting the catch trial to be the same as the cued image) (Figure 3 —figure supplement 5). These values are then averaged to get a priming value where 50% indicates no bias, while higher values indicate a bais towards reporting the mock trails as being the same as the imagined image, while negative numbers indicate a bias towards reporting the oppoiste image to that which was imagined.”

– I would also suggest estimating the null hypothesis and significance bound through random sampling of the experiment under the null to show that the aphantasia subjects did not perform significantly worse than chance in Figure 2b.

We have now run one-sample Bayesian t-tests to assess the evidence for aphantasic individuals performing significantly below chance in both the imaginary pupillary light response and binocular rivalry task (figure 1D). When assessing both groups there was no significant evidence that their priming scores were significantly lower than chance (Comparing scores to 50%: Aphantasic individuals BF = .162, Controls BF = .012). Similarly, a one sample t-test found no significant evidence that either group’s pupil difference scores were lower than chance (comparing to 0: Aphantasic individuals SS1 BF = .860, Aphantasic individuals SS4 BF = .187, Controls SS1 BF = .091, Controls SS4 BF = .050). These analysis have been added to figure 2B as Figure supplement 6 and the following has been added to the discussion:

“We ran Bayesian one-sample t-tests on the binocular rivalry and pupillary light response difference scores (see figure 2) comparing their performance to chance to see if there was any evidence they were performing significantly below chance. We found no significant evidence of below chance performance for either group on either the binocular rivalry or pupillometry imagery tasks (see Figure 2 —figure supplement 6). Taken together, with the set-size pupillary effect we observed in our aphantasic participants, it seems unlikely that our aphantasic individuals were not engaging in the tasks.”

Lines 124 – 127 "These data provide novel evidence individuals can reliably evaluate the comparative vividness of single episodes of imagery. Further, these data demonstrate that the pupillary light response also tracks the phenomenological vividness of visual imagery from moment to moment." These two lines are circular. The authors are using changes in pupil size as evidence that subjects can reliably report their vividness and then using subject reports as evidence that pupil size is a good indicator of vividness. Suggest rewriting /rewording

We have now removed the first line from this paragraph (see manuscript).

Around line 256: the authors state there is no correlation between perception and imagery conditions. I initially found this confusing because if imagery is like faint vision, the pupil change during imagery of an object should mimic the pupil change during viewing of the same object. The point I believe they were making is that there is no correlation when all subjects are pooled together. I think this would be clearer if the authors first pointed out that there is a correlation between perception and imagery, but in the general population only (if true).

The reviewer is correct, that there is no correlation at a group-level between pupil responses during perception, specifically when looking at the control population. Having re-read this section we can see the confusion, because the point of the study is to show that imagery acts like weak perception by showing a pupillary light response. To clarify it was important to run this analysis as it might just be the case that pupil responses during imagery passively reflect the amount the pupils responded during the perception phase, rather than reflecting the effortful generation and maintenance of mental images. Specifically, imagery does act like perception through demonstrating a pupillary light reflex, however this analysis shows that this effect is not just due to lingering pupil responses to the previously seen images. We have added in further clarification of this point to the manuscript and we hope this distinction is now more readily understandable from the added text:

“Another possible explanation of our findings could be that the passive viewing of the images, and lingering visual persistence and sluggish pupil responses could be driving our results. If this is the case, we would expect that pupil diameter during the perception of the images should correlate with pupil size during imagery for the corresponding images. Further, the pupillary light reflex during perception should be more pronounced in the control than the aphantasic populations. To investigate this possible alternative explanation of our data we first assessed the correlations between pupil diameter during perception of bright and dark images for set size one and four and their corresponding imagery conditions (control participants only). We found there were no significant correlations between any of the perception and imagery conditions, or the difference scores for set size one and four (all p >.40, see Figure 2 —figure supplement 1). This lack of a correlation suggests that those individuals who have the largest pupillary light response while viewing the images, do not also have the greatest imagery driven pupillary light responses, making it unlikely that the pupil response while seeing the image is driving the mental imagery pupillary response.”

Line 312 – I believe the authors might be conflating psychogenic and fabrication and suggest that they revise the discussion to be clearer on this. The results speak to the latter but not so much the former. While psychogenic implies there might not be a traceable physical cause, it doesn't necessitate that the patient is acting intentionally and they likely experience it as real (for example, in non-epileptic seizures the brain is in fact not seizing, but the patient often isn't consciously convulsing and truly believes they are having a seizure). The lack of experience of imagery could lead to a lack of pupillary light response. A psychogenic source would not be inconsistent with the findings since the authors are not claiming to identify what is blocking or restricting the visual experience of imagery, just that there is a link between that experience and a physiological response. I suggest removing the claim that the study rules out a psychogenic source to aphantasia.

We thank the reviewer for their thoughtful response, we have now removed this claim in the discussion.

The timing diagram in Figure 1A seems off. The caption and methods say that the black screen rest was presented for 8s, but only 7s exist between the 2nd and 3rd dotted lines, and this doesn't match Figure 3. The reason for using a black screen for rest was also not clear. Together, these made the large dip in pupil size before imagery a bit confusing at first.

Thank you for noting this error in Figure 1A. This has now been amended to correctly show an 8s black screen rest period. The reason for using a black screen for the rest period was firstly, to accurately replicate the experimental design used in Laeng and Sulutvedt, 2014 (The Eye Pupil Adjusts to Imaginary Light). This black screen is included to be a wash out period for after-images caused by stimuli during the perception phase of the trial. Additionally this black screen is used to bring the pupils to a similar diameter so that they are at a similar size for the beginning of the imagery component of the task.